# *BnKAT2* Positively Regulates the Main Inflorescence Length and Silique Number in *Brassica napus* by Regulating the Auxin and Cytokinin Signaling Pathways

**DOI:** 10.3390/plants11131679

**Published:** 2022-06-24

**Authors:** Dashuang Yuan, Yin Zhang, Zhen Wang, Cunmin Qu, Dongming Zhu, Huafang Wan, Ying Liang

**Affiliations:** 1College of Agronomy and Biotechnology, Southwest University, Chongqing 400715, China; yds5206@email.swu.edu.cn (D.Y.); wangzhencq@swu.edu.cn (Z.W.); drqucunmin@swu.edu.cn (C.Q.); dm1361613848@email.swu.edu.cn (D.Z.); 2Academy of Agricultural Sciences, Southwest University, Chongqing 400715, China; 3Chongqing Engineering Research Center for Rapeseed, Chongqing 400715, China; 4Guizhou Rapeseed Institute, Guizhou Academy of Agricultural Sciences, Guiyang 550018, China; zhangyin9412@163.com

**Keywords:** *Brassica napus*, *BnKAT2*, length of main inflorescence, silique number

## Abstract

*Brassica napus* is the dominant oil crop cultivated in China for its high quality and high yield. The length of the main inflorescence and the number of siliques produced are important traits contributing to rapeseed yield. Therefore, studying genes related to main inflorescence and silique number is beneficial to increase rapeseed yield. Herein, we focused on the effects of *BnKAT2* on the main inflorescence length and silique number in *B. napus*. We explored the mechanism of *BnKAT2* increasing the effective length of main inflorescence and the number of siliques through bioinformatics analysis, transgenic technology, and transcriptome sequencing analysis. The full *BnKAT2*(*BnaA01g09060D*) sequence is 3674 bp, while its open reading frame is 2055 bp, and the encoded protein comprises 684 amino acids. *BnKAT2* is predicted to possess two structural domains, namely KHA and _C_NMP-binding domains. The overexpression of *BnKAT2* effectively increased the length of the main inflorescence and the number of siliques in *B. napus*, as well as in transgenic *Arabidopsis thaliana*. The type-A *Arabidopsis response regulator* (*A-ARR*), negative regulators of the cytokinin, are downregulated in the *BnKAT2*-overexpressing lines. The *Aux/IAA*, key genes in auxin signaling pathways, are downregulated in the *BnKAT2*-overexpressing lines. These results indicate that *BnKAT2* might regulate the effective length of the main inflorescence and the number of siliques through the auxin and cytokinin signaling pathways. Our study provides a new potential function gene responsible for improvement of main inflorescence length and silique number, as well as a candidate gene for developing markers used in MAS (marker-assisted selection) breeding to improve rapeseed yield.

## 1. Introduction

Rapeseed (*Brassica napus*) is an important allopolyploid crop formed by interspecific hybridization between *Brassica rapa* and *Brassica oleracea*. It is the second most important oilseed, responsible for approximately 13% of vegetable oil in the world [1,2]. Improving rapeseed yield is an important way to mitigate the disparity between the limited supply and increasing demand, and has been the focus of breeding programs. Three major factors, including the number of siliques, number of seeds per silique, and thousand-seed weight, determine the yield of rapeseed plants, with the number of siliques having the highest correlation with yield [3]. Additionally, an optimized plant architecture (PA) is fundamental for high-yield breeding but the corresponding genetic control is largely unknown in rapeseed. The main inflorescence length is an important trait related to ideal plant architecture [4]. The plant with longer main inflorescence often has higher plant height, more siliques in main inflorescence, and higher yield. Considering current rapeseed production requirements for mechanical sowing with high planting density, the germplasm with longer main inflorescence is very important [5]. When the cultivation density is as high as 75 plants per m^2^, the yield of the main inflorescence accounts for 73.05% of the total yield per plant [6]. Therefore, increasing the effective length of the main inflorescence and the number of siliques may be effectively beneficial to increase the yield [7,8].

The discovery and utilization of rapeseed genes regulating the main inflorescence effective length and silique number is of great importance for breeding plants suitable for mechanized harvesting. Ren et al. [9] detected 17 SNP loci associated with inflorescence length and silique number using genome-wide association study (GWAS) and predicted the candidate genes. Among them, *BnaA01g16940D* encodes a proline-rich glycoprotein that regulates inflorescence length and silique number [9,10,11]. Another gene, *BnaC01g39480D*, encodes phosphoinositide-specific phospholipase C (PI-PLC), which catalyzes the hydrolysis of phosphatidylinositol 4,5-bisphosphate to 1,4,5-triphosphate inositol and diacylglycerol, which are involved in auxin biosynthesis and signal transduction, and ultimately regulates silique number [9,12,13]. Han et al. [14] found that *BnaC06g27150D* affects rapeseed inflorescence structure and stem branching via the auxin and cytokinin signal transduction [15,16,17]. We conducted a GWAS analysis on the effective length of the main inflorescence and the number of siliques in the main inflorescence employing 520 rapeseed cultivars (grown in Lincang, Yunnan, China, and Beibei, Chongqing, China) and screened a candidate gene, namely *BnaA01g09060D*, which might be associated with the length of the main inflorescence and the number of siliques it produces (unpublished). This gene was named *BnKAT2* because of its homology with *At4G18290* (*AtKAT2*) in *Arabidopsis thaliana*. *KAT2* is a potassium channel in plants [18], and in *A. thaliana* it is activated by *AtCPK6* in a calcium-dependent manner to control the K^+^ homeostasis in the leaves [19]. In this study, the genetic sequence of *BnKAT2* and the structural characteristics of its encoded protein were analyzed. The phenotype analysis of transgenic *B. napus* plants and *A. thaliana* plants, both overexpressed with *BnKAT2*, showed that this gene could significantly increase the effective length of the main inflorescence and the number of siliques produced. RNA sequencing (RNA-seq) and differential gene expression analysis results showed that *BnKAT2* functions by regulating the expression of genes that negatively regulate the cytokinin and auxin signal transduction pathways. These findings provide a theoretical basis and germplasm resources for the molecular breeding of rapeseed with a longer main inflorescence and more siliques to improve the yield.

## 2. Results

### 2.1. Cloning and Bioinformatics Analysis of BnKAT2 in B. napus

The full *BnKAT2* gene is 3674 bp in length with a 2055-bp open reading frame and encodes a protein with 684 amino acid residues. The phylogenetic analysis showed that *BnKAT2* is homologous to *Bra013296* in *B. rapa* and to *At4g18290* (*AtKAT2*) in *A. thaliana* (Figure 1a). The BnKAT2 protein was predicted to possess two domains, namely the potassium ion channel KHA domain and the cyclic nucleotide cAMP (_C_NMP)-binding domain (Figure 1c). The _C_NMP-binding domain consists of three α helixes and an eight-chain antiparallel β barrel-like structure. The BnKAT2 protein has four transmembrane helixes distributing at the positions of A64–A86, A101–A123, A201–A220, and A278–A300 (Figure 1b), and the proportions of α helix, β folding, and irregular crimp structures in its secondary structure (Figure 1d) are 50.29%, 9.65%, and 40.06%, respectively.

### 2.2. Improvement Effects of BnKAT2 on the Effective Inflorescence Length and Silique Number

*BnKAT2* was overexpressed in *B. napus* and *A. thaliana* to assess its effect on inflorescence length and silique number. The primers F35S3ND+OEBnKAT2 and Bar-F+Bar-R were used to test the expression of *BnKAT2* in the transgenic plants. As shown in Figure 2a, the expression level of *BnKAT2* was 3.57–5.25 times higher in the transgenic lines overexpressing *BnKAT2* than that in the wild-type *B. napus* (ZS11). Similarly, the expression level of *BnKAT2* was 4.25–5.0 times higher in the transgenic lines than that in wild-type *A. thaliana* (Figure 2b).

The average effective inflorescence lengths of the *BnKAT2*-overexpressing transgenic *B. napus* lines and the wild type were 67.4 and 55.3 cm, respectively, with 21.8% longer inflorescences in the transgenic lines (Figure 3a). The average inflorescence silique numbers in the transgenic *B. napus* lines and the wild type were 95.8 and 71.3, respectively, increasing 34.3% in the transgenic lines (Figure 3a). These results indicate that *BnKAT2* positively regulates the effective inflorescence length and silique number of *B. napus*.

The effective inflorescence length of the transgenic *A. thaliana* was 22.4 cm on average, which was 31.2% longer than that of the wild type (17.1 cm) (Figure 3b). The numbers of siliques produced by the main inflorescences of the transgenic and wild-type *A. thaliana* were 24.3 and 17.9 on average, respectively, increasing 37.3% in the transgenic plants (Figure 3b). This result is consistent with that in *B. napus,* which indicates the pathway may be conserved in *A. thaliana* and *B. napus*.

### 2.3. Analysis of Differentially Expressed Genes

An RNA-seq analysis was performed on the *BnKAT2*-overexpressing and wild-type *B. napus* plants, with 54.56 Gb of clean bases obtained after the quality control process. The proportion of reads with a false discovery rate (FDR) < 0.001 was 93.52–94.46%, and the GC content was over 46.99%, indicating that high-quality transcriptome sequencing data were obtained. A total of 4362 differentially expressed genes (DEGs) were identified between the *BnKAT2*-overexpressing and wild-type *B. napus*, including 1952 upregulated genes and 2410 downregulated genes (Figure 4a). A Venn diagram (Figure 4b) showed that 1587 genes were only expressed in the transgenic *B. napus* lines, while 2016 genes were only expressed in the wild type. These results indicate that *BnKAT2* regulates the effective inflorescence length and silique number by influencing the expression of related genes.

### 2.4. GO and KEGG Enrichment Analysis on the DEGs

To explore the regulating mechanism of *BnKAT2* in details, we completed the enrichment of differential genes employing GO and KEGG analysis. Gene ontology (GO) enrichment analysis was performed using |log2 fold| > 1 and *p* < 0.05 as the threshold for significant enrichment. According to their enrichment significance (*p* < 0.05), the top five GO enrichments in each category were selected for mapping (Figure 5a). In terms of the cell component functions, the major DEG categories have functions associated with the plasma membrane and the nuclear membrane. Among the molecular functions, the DEGs were mainly associated with calcium ion binding and protein binding. The DEGs associated with biological processes were mainly enriched into the intracellular signal transduction pathway and phosphorylated signal transduction pathway categories, which accounted for 1.79% and 1.64% of the DEGs, respectively, suggesting that *BnKAT2* may regulate the effective inflorescence length and silique number via these pathways.

KEGG metabolic pathway analysis showed that the upregulated and downregulated DEGs were classified in 75 and 89 metabolic pathways, respectively. According to their enrichment significance (*p* < 0.05), the top 10 KEGG pathways were selected for mapping (Figure 5b). The DEGs were mainly enriched in plant hormone signal transduction. Among them, 24 DEGs were enriched in plant hormone signaling pathways (Appendix A), with 18 of the genes downregulated in the *BnKAT2*-overexpressing plants found to be associated with the auxin and cytokinin signaling pathways (Figure 5c). The downregulated genes in the auxin signaling pathway belong to the Aux/IAA gene family, a negative regulator of the auxin signaling pathway. The downregulated genes in the cytokinin signaling pathway are mainly *AHP6* and type-*A*
*response regulator* (A-ARR) genes, which are negative feedback regulatory genes in the cytokinin signaling pathway. These results suggest that *BnKAT2* may affect the effective inflorescence length and silique number through the regulation of auxin and cytokinin signal transduction.

### 2.5. DEGs Verification Using qRT-PCR

To verify that *BnKAT2* is involved in the cytokinin and auxin signaling pathways, the expression level of key genes involved in these pathways that were significantly downregulated in the RNA-Seq data of *BnKAT2*-overexpressing plants were quantified using qRT-PCR. The cytokinin signaling-related genes, including *BnaA06g06240D* (*ARR4*), *BnaC08g27970D* (*ARR9*), *BnaA06g22370D* (*ARR6*), *BnaA03g19410D* (*ARR3*), and *BnaC05g14720D* (*ARR7*), were confirmed to be significantly downregulated in the *BnKAT2*-overexpressing plants compared to the wild type (Figure 6), as were the auxin signaling-related genes *BnaCnng41350D* (*IAA1*) and *BnaA09g16590D* (*IAA2*). Compared with the expression level in the wild type, the expression levels of the *ARR* genes in the *BnKAT2*-overexpressing plants ranged from 1.41 to 20.51, on average, while in the wild type, the expression levels ranged from 6.10 to 40.49. The expression of *ARR4* showed the highest decrease in expression (83.6%) in the *BnKAT2*-expressing plants. The expression levels of *IAA1* and *IAA2* were 0.23 and 0.46, on average, in the *BnKAT2*-overexpressing plants and 2.42 and 2.7 in the wild type, representing a decrease of 21.8% and 58.7%, respectively. The expression patterns of the DEGs determined using qRT-PCR were therefore consistent with the RNA-seq results, confirming the reliability of RNA-Seq data.

## 3. Discussion

Seed yield, one of the most important goal for rapeseed breeding, is directly determined by silique number, seeds amount, and seed weight [20]. The effective length of the main inflorescence and the number of siliques it produces are closely correlated with rapeseed yield [21]. In a previous GWAS analysis (unpublished), *BnKAT2* was identified as a putative regulating gene responsible for the indices mentioned above, but this had not been verified and the corresponding mechanism had not been unraveled.

In the present study, the *BnKAT2*-overexpressing *B. napus* and transgenic *BnKAT2*-expressing *A. thaliana* lines produced longer inflorescences and more siliques than their respective wild types. The BnKAT2 protein was predicted to contain a cNMP (cAMP and cGMP) binding domain, indicating that *BnKAT2* may participate in intracellular signal transduction pathway. RNA-seq was performed on the *BnKAT2*-overexpressing and wild-type *B. napus* plants, and GO enrichment analysis showed that the DEGs between these genotypes were enriched in cellular signal transduction pathways. KEGG enrichment analysis showed that the DEGs were mainly enriched in the auxin and cytokinin signaling pathways.

The main DEGs in auxin pathway belonged to the Aux/IAA family, which plays key roles in the auxin response [22,23,24]. Auxin is essential for the elongation of plant stems [25,26], and for the initiation of floral meristems (FMs) as well as termination meristem growth [27]. However, the regulating mechanism on the floral organ initiation remains less mechanistically well understood than its functions in the development of other organs. Auxin can also function as a morphogen by generating response gradients, particularly in combination with the cytokinin, which are instrumental in several aspects of floral organogenesis [28]. Auxin regulates flower development in *A. thaliana* involving members of the AINTEGUMENTA-LIKE/PLETHORA(AIL/PLT) gene family [29]. IAA can activate guanosine cyclase, leading to the increase of endogenous cGMP in *A. thaliana* roots and the degradation of Aux/IAA proteins by cGMP-dependent protein kinases. This promotes the expression of the *IAA*-responsive genes and ultimately affects auxin signal transduction [30,31,32]. In the present study, the *Aux/IAA* genes (*IAA1*, *IAA2*) were downregulated in the *BnKAT2*-overexpressing *B. napus* plants, probably due to the activation of cGMP following its binding to the _C_NMP-binding domain in *BnKAT2* proteins, which also caused the degradation of Aux/IAA protein. Thus, the inhibitory effect of the Aux/IAA proteins on the expression of the genes downstream of the auxin signaling pathway was lifted, leading to their upregulated expression, and the length of main inflorescence and the number of siliques were increased. The specific mechanism underlying this effect is yet to be fully elucidated.

It showed that, in the *BnKAT2*-overexpressing *B. napus* line, eight of the downregulated genes were involved in the cytokinin signaling pathway, seven of which belonged to the A-ARR gene family and one was *AHP6* (Table 1). The *ARR* gene family comprises regulatory proteins that function in response to cytokinin. A-type ARRs are a family of primary cytokinin response genes, which interfere with cytokinin activity and encode phosphorus-accepting proteins of the two-component response regulator class [33,34]. WUSCHEL (WUS), a positive regulator of stem cells, directly represses the transcription of several two-component sensor-regulator *ARR* genes (*ARR5*, *ARR6*, *ARR7*, and *ARR15*), which act in the negative feedback loop in cytokinin signaling, suggesting that the *ARR* genes might negatively influence meristem size and that their repression by WUS might be necessary for meristem function [35]. Loss-of-A-type ARR function in *ARR7* and *ARR3*,*4*,*5*,*6*,*7*,*8*,*9* mutants strongly stimulated callus development, indicating that cell proliferation is repressed by A-type ARRs [36]. Loss of ARR activity in maize causes enlarged meristems [37]. *AHP6* simultaneously inhibits the transfer of phosphate from *AHP1* to *ARR1* and the transfer from histidine to aspartic acid in the AHKs, suggesting that *AHP6* may be a negative regulator of cytokinin signal transduction [38]. Our results suggest that *BnKAT2* is involved in cytokinin signal transduction by negatively regulating the expression of *A-ARR* genes and *AHP6*. Foliar spraying of cytokinins can improve rapeseed yields by significantly increasing the number of siliques per plant, stimulating the fruit-bearing characteristics of the main inflorescence, and reducing the seed shattering rate [39]. Increased epidermal cytokinin biosynthesis or breakdown could be achieved through the expression of the cytokinin biosynthesis gene *LOG4* or the cytokinin-degrading gene *CKX1*, respectively, under the control of the epidermis-specific *AtML1* promoter in *A. thaliana* [40]. During vegetative growth, increased epidermal cytokinin production led to larger shoot apical meristems and promoted earlier flowering [41]. Auxin negatively regulates cytokinin biosynthesis through the iPMP-independent cytokinin biosynthesis pathway, and cytokinin and auxin act synergistically during organ initiation to jointly regulate the growth of the shoot apical meristem [42]. We speculate that *BnKAT2* is involved in auxin signaling by regulating the expression of the *Aux/IAA* gene via binding to cGMP and in the cytokinin signaling pathway by regulating the expression of the *ARR* and *AHP* genes. Through these network pathways, *BnKAT2* regulates the growth of the main inflorescence and the number of siliques in *B. napus.* A possible model for this *BnKAT2* regulatory pathway is presented in Figure 7. However, the mechanism underlying the integration of *BnKAT2* with cytokinin and auxin signaling pathways has not been fully elucidated yet.

We also noticed that *BnKAT2* and *AtKAT2* protein domains contain a potassium channel domain (KHA). Potassium is involved in turgor-driven processes such as cell elongation, phototropism, gravitropism, and stomatal movement [43]. The activity of K^+^ channels has been shown to be dependent on auxin via the activation of an auxin-binding protein and the induction of cytosolic pH changes [44,45]. Auxin-stimulated proton efflux and potassium uptake into expanding coleoptile cells affect the upregulation of the H^+^-ATPase [46,47] and activation of K^+^ uptake [48]. Re-addition of K^+^ or removal of K^+^ channel blockers in the presence of auxin leads to a rapid enhancement of coleoptile growth rates [49]. Auxin-induced growth in maize coleoptile segments involves K^+^ uptake through voltage-dependent, inwardly rectifying K^+^ channels (*ZMK1*, *Zea mays* K^+^ channel 1), the activity of which contributes to water uptake and consequently influences cell expansion [50]. The target of auxin action is the plasma membrane H^+^-ATPase, which excretes H^+^ into the cell wall compartment and, in an antiport, takes up K^+^ through an inwardly rectifying K^+^ channel. The auxin-enhanced H^+^ pumping lowers pH value in the cell wall, resulting in the activation of pH-sensitive enzymes and proteins within the wall, and cell wall loosening and extension growth [46,47,50]. It is speculated that auxin activates the K^+^ channel (KHA) of BnKAT2 protein through excreting H^+^ and stimulating K^+^ influx, which eventually leads to cell elongation and growth and increases the effective length of the main inflorescence.

Rapeseed siliques are not only a storage organ, but also an important photosynthetic organ, and thus play dual roles of “source” and “sink”, which further influences yields [51]. *BnKAT2* affects the length of the main inflorescence and the number of siliques produced by *B. napus,* both of which are important factors affecting the yield. It can be employed as a target to develop markers for screening out the germplasm source for yield breeding in *B. napus*.

It is noteworthy that effective silique number on main inflorescence is obviously affected by the planting environment [21,52]. The yield of crop is always affected by abiotic stress, such as drought and so on [21,53]. The response of *BnKAT2* to stress possibly be very important for because of the influence on the vegetative and reproductive growth of plants. *BnKAT2* improves the main inflorescence length and the number of siliques under the suitable cultivation environment in the present research, which is meaningful to unravel the response mechanism to improve the yield. This study will aid theoretical research on improving rapeseed yields and also provides a new target gene for the improvement of important crops through genetic engineering. However, the effect of *BnKAT2* expression on possible reproductive problems such as lack of seeds setting or others, which play an important role in yield improvement, has not been investigated in the present study.

## 4. Materials and Methods

### 4.1. Plant Materials and Growing Conditions

*B**. napus* (ZS11) and *A. thaliana* (Col-0) were used in this study. The *B. napus* plants were grown in the experimental field of Chongqing Rapeseed Engineering Technology Research Center (29°49′08′′ N, 106°25′11′′ E; altitude of 216 m) and were cultivated and managed using conventional procedures. The *A. thaliana* plants were cultured in growth chamber, as described by Wang [54].

### 4.2. Cloning of BnKAT2 in B. napus

Appropriate *B. napus* leaves were harvested to extract a high-quality RNA using an RNA prep Pure Plant Kit (Tiangen, Beijing, China). The cDNA was reverse-transcribed according to the PrimeScript RT Reagent Kit with a gDNA Eraser (Perfect Real Time) kit (Takara Bio, Kusatsu, Japan). *BnKAT2*-specific primers (Table 1) were designed according to the genetic sequence on the NCBI website (http://www.ncbi.nlm.nih.gov/ (accessed on 2 May 2021)). Using the total *B. napus* cDNA as template, *BnKAT2* was amplified and ligated into a pGEM-T easy vector and then was transformed into *Escherichia coli* DH5α cells. The positive clones were screened out and sequenced by Shenzhen Huada Gene Company (Shenzhen, China) to obtain the full-length *BnKAT2* cDNA sequence.

### 4.3. Bioinformatics Analysis

PredictProtein and TMHMM Server v.2.0 (https://services.healthtech.dtu.dk/service.php?TMHMM-2.0 (accessed on 2 November 2020)) were used to predict the secondary structures and transmembrane domains of the BnMAPK2 protein. MEGA11.0 software (https://www.megasoftware.net/ (accessed on 2 November 2020)) was employed to align the amino acid sequences and construct the phylogenetic tree of BnKAT2. The BnKAT2 protein domains were predicted using the online software on the Prosite website (http://prosite.expasy.org/prosite.html (accessed on 2 November 2020)).

### 4.4. Plasmid Construction and Plant Transformation

To generate the *BnKAT2* transgenic expression vector, the *BnKAT2* fragment was amplified from the recombinant vector PGEM-T-*BnKAT2*. For the *A. thaliana* transformation, the *BnKAT2* fragment was cloned into Pearleygate101 to make a Pearleygate101-35S-*BnKAT2* construct, which was transferred into *Agrobacterium tumefaciens* strain GV3101 and transformed into *A. thaliana* Col-0 plants using the floral dip method [55]. Transgenic lines were selected on 0.5× Murashige and Skoog medium containing hygromycin (100 mg/L). For the *B. napus* transformation, the *BnKAT2* construct was introduced into the cotyledons of *B. napus* using an *Agrobacterium*-mediated method [56]. The PCR primers used in these experiments are listed in Table 1.

### 4.5. Phenotypic Examination of the Transgenic Plants

The effective length of the main inflorescence (from the first silique to the last silique in the main inflorescence) and the number of siliques produced on the main inflorescence were measured at the maturity stage of the transgenic and wild-type *B. napus* and *A. thaliana* plants. The data represent three experiments for each genotype, each including genotype replicates of 20 plants.

### 4.6. RNA-Seq

The stem tips of the *BnKAT2*-overexpressing and wild-type *B. napus* plants at the bolting stage were harvested and immediately frozen in liquid nitrogen. The RNA-seq was performed by Beijing Nuohe Source Technology (Beijing, China). The original reads were first converted into sequencing reads using the software CLC Genomics Workbench 9 (Qiagen, Hilden, Germany), and the reads containing ambiguous bases were removed to enhance the quality of the data. The clean data were mapped to the *B. napus* reference genome (http://www.genoscope.cns.fr/brassicanapus/ (accessed on 5 June 2021)). Reference genome indexing and read alignments were performed using Bowtie 2.0.6 and TopHat 2.0.9 (https://magic.novogene.com/ (accessed on 5 June 2021)), respectively. The number of reads per kilobase per million mapped reads (RPKM) [57] was used to calculate the relative levels of expression. Genes with a |log2FC| > 1 and a *p*-value < 0.05 were considered to be differentially expressed between the two genotypes. The GOseq package (https://magic.novogene.com/ (accessed on 5 June 2021)) was used to perform a GO enrichment analysis of the DEGs. The GO terms with a corrected *p*-value < 0.05 were considered significantly enriched [58]. The KEGG statistical enrichment of the DEGs was evaluated using KOBAS software (http://kobas.cbi.pku.edu.cn (accessed on 5 June 2021)). The DEGs were grouped into the KEGG pathways based on their matches to the database [59].

### 4.7. QRT-PCR Analysis

The qRT-PCR analysis was performed on seven of the identified DEGs to test the reliability of the transcriptome sequencing data and verify the effects of *BnKAT2* on the expression of genes negatively regulating the cytokinin and auxin signaling pathways. The reverse transcription was performed using a PrimeScript RT Reagent Kit (Takara Bio). The qRT-PCR was performed on a Light Cycler 480 II (Roche, Basel, Switzerland) using SYBG Premix Ex Taq II (Takara Bio), with the following cycling conditions: 95 °C for 30 s; followed by 40 cycles of 95 °C for 5 s, 60 °C for 30 s, and 60 °C for 45 s. Actin7 was used as the reference gene. The primer sequences specific to the 12 DEGs are shown in Appendix A. The relative gene expression levels were calculated according to 2^−ΔΔCT^ [60].

## 5. Conclusions

*BnKAT2* was highly homogenous to the *AtKAT2* (*AT4G18290*) gene in *A. thaliana*. The BnKAT2 protein has KHA and _C_NMP-binding domains and is speculated to participate in the auxin signaling pathway by regulating *Aux/IAA* gene expression via targeting cGMP. It also participates in the cytokinin signaling pathway by regulating the ARR gene. It integrates the auxin and cytokinin signaling pathways to regulate the effective length of the main inflorescence of *B. napus*, as well as the number of siliques produced.

## Figures and Tables

**Figure 1 plants-11-01679-f001:**
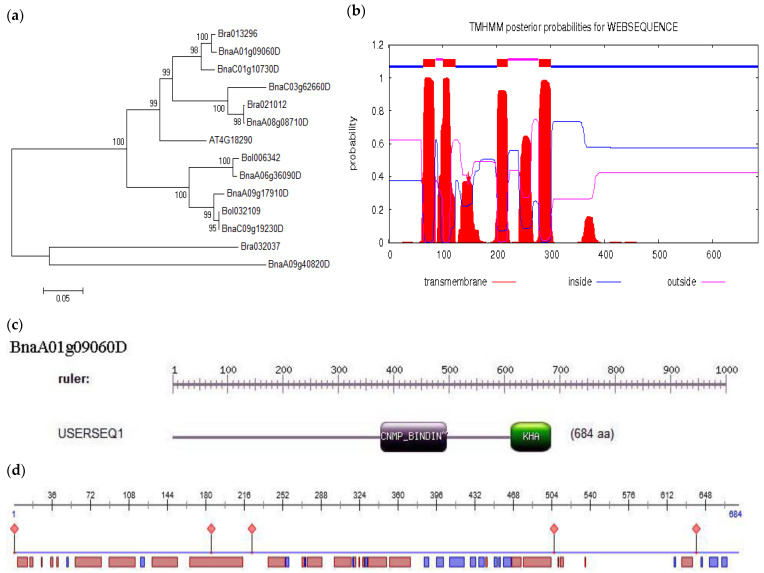
Bioinformatics analysis of the *BnKAT2* gene. (**a**) Phylogenetic tree. (**b**) Transmembrane structure of the BnKAT2 protein. The blue line, thick red line, and pink line represent the area inside the membrane (A1–A63; A124–A200; A301–A684), the transmembrane region (A64–A86; A101–A123; A201–A220; A278–A300), and the area outside the membrane (A87–A100; A221–A277), respectively. (**c**) Protein domains of BnKAT2 (BnaA01g09060D). (**d**) Secondary structure of the BnKAT2 protein. The red rectangle indicates the α-helix, the blue rectangle indicates the β-sheet, and the rest of the sequence is random coils; the red rhombus indicates a protein binding site.

**Figure 2 plants-11-01679-f002:**
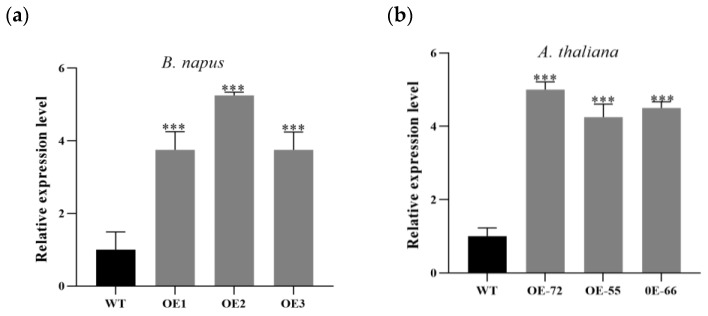
The expression of *BnKAT2* in transgenic *B. napus* (**a**) and *A. thaliana* (**b**)**,** relative to their respective wild-type expression levels. Values are means ± SD of three independent biological replicates. *** indicate significant differences at the 0.001 levels.

**Figure 3 plants-11-01679-f003:**
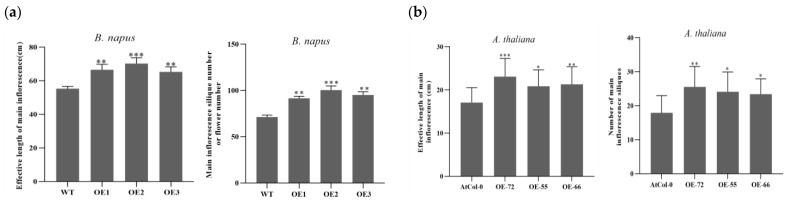
Comparison of the effective main inflorescence length and number of siliques produced by the transgenic and wild-type *B. napus* (**a**) and *A. thaliana* (**b**) plants. Data are means ± SD (*n* = 3 biological replicates of 20 plants per genotype). *, **, and *** indicate significant differences at the 0.05, 0.01, and 0.001 levels, respectively.

**Figure 4 plants-11-01679-f004:**
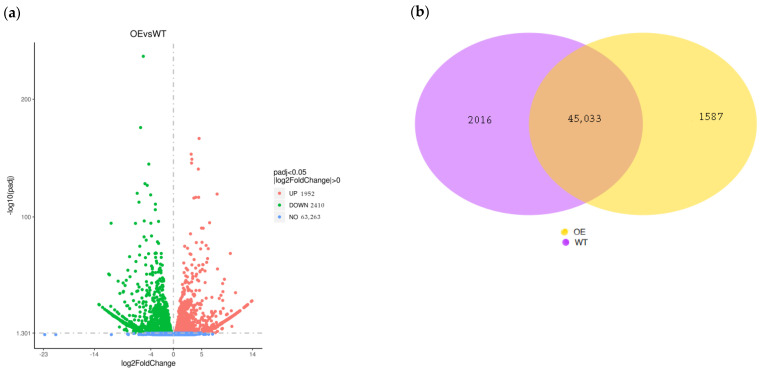
Volcano (**a**) and Venn diagram (**b**) of the significant DEGs between the *BnKAT2*-overexpressing (OE) and wild-type (WT) *B. napus*.

**Figure 5 plants-11-01679-f005:**
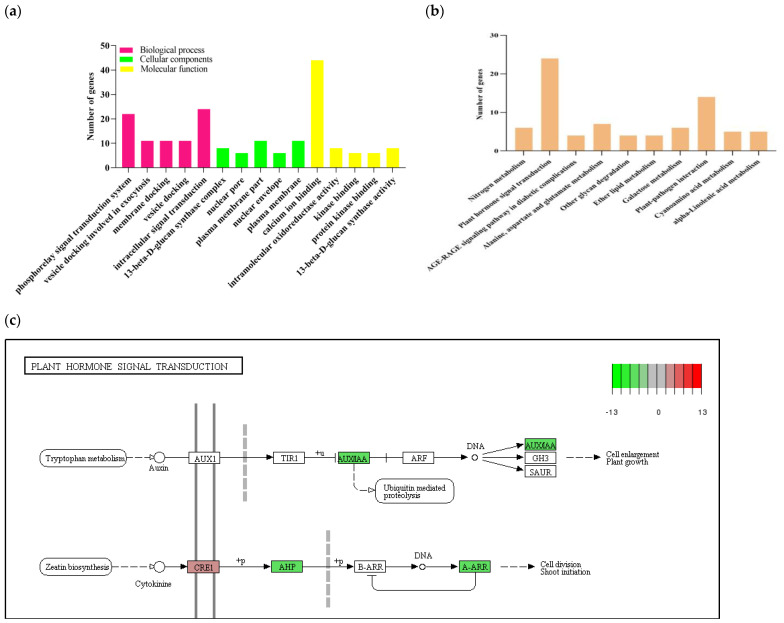
Functional analysis of the differentially expressed genes (DEGs) between the *BnKAT2*-overexpressing and wild-type *B. napus* plants. (**a**) GO enrichment of DEGs. (**b**) KEGG enrichment of DEGs. (**c**) Phytohormone signaling pathways associated with the genes downregulated (green) and upregulated (red) in the *BnKAT2*-overexpressing plants.

**Figure 6 plants-11-01679-f006:**
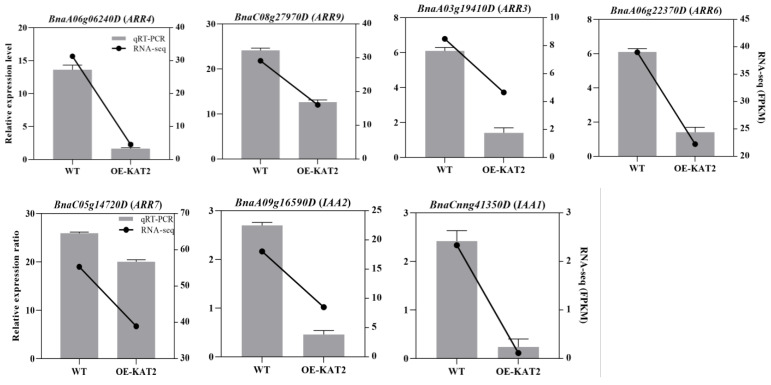
qRT-PCR verification of some of the DEGs identified in the RNA-seq analysis. The relative expression values for the RNA-seq data are the average fragments per kilobase of exon per million mapped fragments (FPKM) values of three biological replicates. The qRT-PCR value represents the mean relative expression values. Values suggest means ± SD of three independent biological replicates.

**Figure 7 plants-11-01679-f007:**
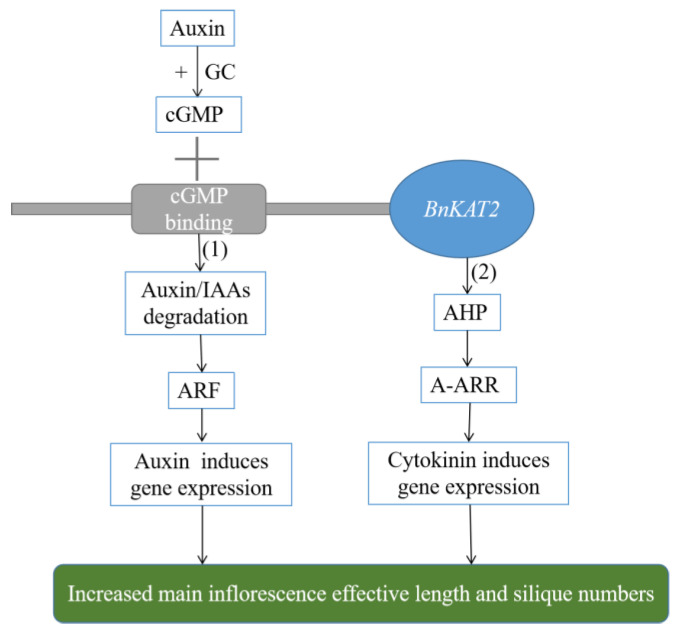
Regulation model of *BnKAT2* on the effective length of the main inflorescence and number of siliques in *B. napus.* (**1**) *BnKAT2* participates in the auxin signaling pathway by regulating the expression of *Aux/IAA* by cGMP. (**2**) *BnKAT2* participates in the auxin signaling pathway by regulating the expression of *AHP* and *ARR* genes.

**Table 1 plants-11-01679-t001:** Primers used in this study.

Primer Name	Forward Primer (5′–3′)	Reverse Primer (5′–3′)
BnKAT2	ATGTCAATCTCTTGCACTAGAAACTTCTTT	TCAAGAGTCTATGCTTTCAAGCTCAC
BnKAT2q	CAACATTGTCAATGGCTTCTTTGC	TGGAATGGAGCTGTGGAGCAGA
ACT7	TGGGTTTGCTGGTGACGAT	ACGGATCCTAGCAGCTCAGATGTTGA
OE-BnKAT2	CACCATGTCAATCTCTTGCACTAGAAACTTCTTTG	AGAGTCTATGCTTTCAAGCTCACTG
35S3NDF-RPA3ND	GGAAGTTCATTTCATTTGGAGAG	AGAGTCTATGCTTTCAAGCTCACTG
Bar	CGACATCCGCCGTGCCACCGA	CAAATCTCGGTGACGGGCAGG

## Data Availability

Not applicable.

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
