# Peer review of "BnKAT2 Positively Regulates the Main Inflorescence Length and Silique Number in Brassica napus by Regulating the Auxin and Cytokinin Signaling Pathways"

_plants, 2022, doi:10.3390/plants11131679_

Round 1

Reviewer 1 Report

Dear Editor,

Thank you for concerning me as a reviewer of the manuscript submitted to your journal, entitled: "Effects of BnKAT2 on the Main Inflorescence Length and Si-2 lique Number in Brassica napus". The Authors showed that the overexpression of BnKAT2 effectively increased the length of the rapeseed inflorescence and the number of siliques and is involved in the cytokinin and auxin signal transduction pathways. Although the work is interesting, I think that the Authors should take a count of slight modifications of this article. I recommend publishing it in "Plants" after a minor revision.

General remarks:

-          name of species and genus should be written in italic font (e. g. Brassica napus, Arabidopsis thaliana, Agrobacterium rhizogenes etc…)

-          the text of the article needs the thorough editorial correction (e. g. line 89- font color etc.)

Abstract:

Line 16:  Brassica napus is the dominant rapeseed cultivated in China -  is there another rapeseed species?

Keywords:

Authors should take into account that keywords, according to the rules of writing scientific papers, should not be the same as in the title

Introduction

Line 43-44:  ….. therefore, increasing the effective length of the main inflorescence and the number of siliques can effectively increase the yield [4-5]. - assuming that there are no reproductive problems affecting seeds yields

Discussion

The Authors rightly hypothesize that the BnKAT2 target gene could be used for the improvement of important crops through genetic engineering, but not necessarily increasing the size of inflorescences and the number of siliques may translate into increased seeds yield, one should take into account possible reproductive problems, lack of seeds setting etc ....., which of course may be the aim of different study.

With best regards,
Monika Tuleja

Author Response

plants-1753466 manuscript revision instructions

Dear Reviewers:

 Thank you for reading our manuscript (ID: plants-1753466) in your busy schedule and for providing us with such valuable comments and suggestions, which are of great help to our research and writing. We have carefully read your review comments, carefully analyzed and revised the manuscript, and responded to your questions as follows:

1.name of species and genus should be written in italic font (e. g.Brassica napus, Arabidopsis thaliana, Agrobacterium rhizogenesetc…)

We have checked the manuscript in detail and wrote the name of species and genus in italic font.

2.the text of the article needs the thorough editorial correction (e. g. line 89- font color etc.)

We have checked and revised.

3.Line 16:Brassica napusis the dominant rapeseed cultivated in China - is there another rapeseed species?

We revised expression in the manuscript.

4.Authors should take into account that keywords, according to the rules of writing scientific papers, should not be the same as in the title

We have changed the key words according to the suggestions of the reviewer.

5.Line 43-44: …..therefore, increasing the effective length of the main inflorescence and the number of siliques can effectively increase the yield [4-5]. -assuming that there are no reproductive problems affecting seeds yields

As the reviewer mentioned, the reproductive capacity is important for the seed yield. Therefore, we added the “assuming that there are no reproductive problems affecting seeds yields” in the revised manuscripts.

6.The Authors rightly hypothesize that the BnKAT2 target gene could be used for the improvement of important crops through genetic engineering, but not necessarily increasing the size of inflorescences and the number of siliques may translate into increased seeds yield, one should take into account possible reproductive problems, lack of seeds setting etc ....., which of course may be the aim of different study.

As the reviewer’s opinion, the seed yield is affected by many factors including reproductive problems and the size of inflorescences, as well as the number of siliques. However, During plant growth, we only observed that the effective length of the main inflorescence and the number of siliques of the plants overexpressing BnKAT2 were different from those of the wild-type plants. The effective length of the main inflorescence and the number of siliques were one of the factors affecting plant yield, so only Pay attention to the effective length of the main inflorescence and the number of siliques.we will further to investigate the possible effect on the reproductive capacity in the following research.

With best regards,

Ms. Yuan

Reviewer 2 Report

Dear authors,

you are presenting a well done paper with clear structures and intersting results.

Some comments:

1. You may intersted in the follwing paper (Wilmowicz et al. 2009, Int. J. mol. Sci 20 (15) and could include the results into your discussion

2. You may increase the discussion by an additional factor. How important are the results for breeding?

3. What is about the dependence on enviromental factors? Stress facotors like high temperature or drought stress?

Author Response

plants-1753466 manuscript revision instructions

Dear Reviewers:

 Thank you for reading our manuscript (ID: plants-1753466) in your busy schedule and for providing us with such valuable comments and suggestions, which are of great help to our research and writing. We have carefully read your review comments, carefully analyzed and revised the manuscript, and responded to your questions as follows:

1.You may be interested in the following paper (Wilmowicz et al. 2009, Int. J. mol. Sci 20 (15) and could include the results into your discussion.

We studied the paper (Wilmowicz et al. 2019, Int. J. mol. Sci 20 (15)) and discussed the corresponding content in the discussion.

2.You may increase the discussion by an additional factor. How important are the results for breeding?

We studied much more related reference involved in the yield of rapeseed and discussed the application of the results in breeding, which can be found in the revised manuscript.

3.What is about the dependence on environmental factors? Stress factors like high temperature or drought stress?

The environmental factors can affect the seed yield, such as temperature or drought stress. However, we did not investigate the influence in this study. The reviewer offered an important remind. We will perform the associating research involved in the response of the BnKAT2 expression to the abiotic stress, especially in the reproductive capacity. Therefore, we added the corresponding discussion in this manuscript.

Reviewer 3 Report

  1. The title should be revised as the present title is not appropriate
  2. Start the abstract with a clear statement on the scope, relevance, and intention of the study, before describing the main results. End the abstract with a clear statement about the main conclusions and perspectives of the work.
  3. Improve the introduction section with recent relevant studies. Should clearly explain the scope, importance, and incentive of the work in a manner that is understandable by the educated biologist.
  4. Figure 2; Authors should add some statistics to show significance differentiation among the presented values.
  5. Use Italics for Gene and scientific names
  6. Figures 5 and 6 presentations need to be improved.
  7. The quality of the English language and grammar for the entire manuscript needs to improve by native English speakers.

Author Response

plants-1753466 manuscript revision instructions

Dear Reviewers:

 Thank you for reading our manuscript (ID: plants-1753466) in your busy schedule and for providing us with such valuable comments and suggestions, which are of great help to our research and writing. We have carefully read your review comments, carefully analyzed and revised the manuscript, and responded to your questions as follows:

1.The title should be revised as the present title is not appropriate

We have changed the title of the manuscript as “BnKAT2 Positively Regulate the Main Inflorescence Length and Silique Number in Brassica napus via regulating the auxin and cytokinin signaling pathways”. 

2.Start the abstract with a clear statement on the scope, relevance, and intention of the study, before describing the main results. End the abstract with a clear statement about the main conclusions and perspectives of the work.

We carefully revised the abstract according to the reviewer’s suggestions. We added a more clear statement about scope, relevance and incentive of this work before the main results. At the end of the abstract, we added a statement about the conclusions and the corresponding perspectives of the work. 

3.Improve the introduction section with recent relevant studies. Should clearly explain the scope, importance, and incentive of the work in a manner that is understandable by the educated biologist.

We have studied much more reference, and further more clearly explain the scope, importance and incentive of the work.  

4.Figure 2;Authors should add some statistics to show significance differentiation among the presented values.

We added the statistics to show the level of significant difference among the presented values in Figure 2 in the revised manuscript.

5.Use Italics for Gene and scientific names

We thoroughly checked and modified the italics for gene, species and genus in the manuscript.

6.Figures 5 and 6 presentations need to be improved.

We improved the presentations for Figures 5 and 6, which can be found in the revised manuscript.

7.The quality of the English language and grammar for the entire manuscript needs to improve by native English speakers.

We asked an American teacher to help us improve the English language and grammar for the manuscript.
